# Chlorophyll Derivatives from Marine Cyanobacteria with Lipid-Reducing Activities

**DOI:** 10.3390/md17040229

**Published:** 2019-04-17

**Authors:** Sara Freitas, Natália Gonçalves Silva, Maria Lígia Sousa, Tiago Ribeiro, Filipa Rosa, Pedro N. Leão, Vitor Vasconcelos, Mariana Alves Reis, Ralph Urbatzka

**Affiliations:** 1Interdisciplinary Center of Marine and Environmental Research (CIIMAR/CIMAR), University of Porto, Terminal de Cruzeiros de Leixões, Av. General Norton de Matos s/n, 4450-208 Matosinhos, Portugal; sfreitas@ciimar.up.pt (S.F.); nsilva@ciimar.up.pt (N.G.S.); msousa@ciimar.up.pt (M.L.S.); tribeiro@ciimar.up.pt (T.R.); frosa@ciimar.up.pt (F.R.); pleao@ciimar.up.pt (P.N.L.); vmvascon@fc.up.pt (V.V.); mreis@ciimar.up.pt (M.A.R.); 2FCUP, Faculty of Science, Department of Biology, University of Porto, Rua do Campo, Alegre, 4169-007 Porto, Portugal

**Keywords:** zebrafish Nile red fat metabolism assay, anti-obesity drugs, chlorophyll derivatives, murine pre-adipocytes, PPARγ

## Abstract

Marine organisms, particularly cyanobacteria, are important resources for the production of bioactive secondary metabolites for the treatment of human diseases. In this study, a bioassay-guided approach was used to discover metabolites with lipid-reducing activity. Two chlorophyll derivatives were successfully isolated, the previously described 13^2^-hydroxy-pheophytin a (**1**) and the new compound 13^2^-hydroxy-pheofarnesin a (**2**). The structure elucidation of the new compound **2** was established based on one- and two-dimensional (1D and 2D) NMR spectroscopy and mass spectrometry. Compounds **1** and **2** showed significant neutral lipid-reducing activity in the zebrafish Nile red fat metabolism assay after 48 h of exposure with a half maximal effective concentration (EC_50_) of 8.9 ± 0.4 µM for **1** and 15.5 ± 1.3 µM for **2**. Both compounds additionally reduced neutral lipid accumulation in 3T3-L1 multicellular spheroids of murine preadipocytes. Molecular profiling of mRNA expression of some target genes was evaluated for the higher potent compound **1,** which indicated altered peroxisome proliferator activated receptor gamma (PPARγ) mRNA expression. Lipolysis was not affected. Different food materials (*Spirulina*, *Chlorella*, spinach, and cabbage) were evaluated for the presence of **1,** and the cyanobacterium *Spirulina*, with GRAS (generally regarded as safe) status for human consumption, contained high amounts of **1**. In summary, known and novel chlorophyll derivatives were discovered from marine cyanobacteria with relevant lipid-reducing activities, which in the future may be developed into nutraceuticals.

## 1. Introduction

In recent years, the discovery of natural products was extended to new pharmaceutical targets in addition to the traditional targets explored in the past decades, for example, in the search of new anti-obesogenic compounds [1]. Obesity is a complex metabolic disease characterized by an abnormal fat accumulation in adipocytes which are also important regulators of the whole metabolism and homeostasis [2]. This disease is an increasing epidemic, since a considerable percentage of the world’s population is overweight, which is associated with several chronic diseases like diabetes, cardiovascular diseases, and cancer [2,3].

The ineffective pharmacological treatment of current anti-obesity drugs available on the market is due to the limitation of long-term success, potentially dangerous side effects, and high costs [4]. Alternatively, some natural products and derived compounds are being used in the clinic to treat obesity, for example, orlistat, a synthetic derivative of lipostatin isolated from *Streptomyces toxytricini* [5]. Yoshinone A isolated from the marine cyanobacterium *Leptolyngbya* sp., and other marine natural products containing a 7-en-γ-pyrone moiety showed promising anti-obesogenic activity [6].

Although the marine environment is a rich source of new natural products with a high range of applications [7], this environment is still largely underexploited due to inaccessibility and difficulties in retrieving these organisms from their habitat into laboratorial cultures. Cyanobacteria, a group of ancient photoautotrophic microorganisms, are interesting resources for natural product discovery in marine environments [8]. These organisms are known to produce a wide range of bioactive secondary metabolites and most often adapt well to laboratorial culture conditions [9,10]. Our in-house cyanobacterial culture collection (LEGEcc) currently harbors about 500 cyanobacterial strains isolated from the Portuguese coast and other environments that overall represent a largely untapped source for discovery of new secondary metabolites [11]. Some natural compounds were already successfully isolated from this collection, for example, hierridin B [12], bartolosides (A–K) [13,14], and portoamides (A–D) [15], all with cytotoxic activity on cancer cell lines. 

Zebrafish (*Danio rerio*) is an attractive model organism for biomedical research. In the study of complex metabolic disorders like obesity, the use of more complex in vivo model systems as small whole animal models can bring significant advantages. For drug discovery, zebrafish assays can be used complementary to rodent assays with easier handling, high predictive validity and cost-efficiency, while compatible with high-throughput screening [16]. Zebrafish possess higher physiological relevance than cellular in vitro models, more interactions between tissues and have genetic homology to mammals, as well as significant similarities in lipid metabolism [16,17]. The original procedure of the zebrafish Nile red fat metabolism assay was published by Jones et al. [18] and analyzed the capacity of compounds to reduce neutral lipids in zebrafish larvae in vivo. The authors concluded that the zebrafish organism model can be used for identifying non-toxic molecules for treating clinical obesity due the conservation of signal transduction pathways that regulate lipid metabolism.

The aim of this study was to uncover new cyanobacterial compounds with lipid-reducing activity using the zebrafish Nile red fat metabolism assay optimized in our laboratory. We report the isolation and structural elucidation of a known and a novel chlorophyll derivative from the marine cyanobacteria *Cyanobium* sp. LEGE 07175 and *Nodosilinea* sp. LEGE 06001, respectively. Their lipid-reducing activity was additionally evaluated in a three-dimensional (3D) cell culture model of murine pre-adipocytes. Initial molecular profiling was performed for the more potent compound **1**. Furthermore, the presence of **1** was evaluated in different photoautotrophic organisms (*Spirulina*, *Chlorella*, spinach, and cabbage), with GRAS (generally regarded as safe) status for human consumption. 

## 2. Results

### 2.1. Isolation of Compound **1**

Compound **1** was obtained as a green dark amorphous solid from the cyanobacterium *Cyanobium* sp. LEGE 07175. The isolation required several chromatographic steps and was guided by a strong reduction of lipid content observed in the zebrafish Nile red fat metabolism assay. An LC-ESI-LRMS (liquid chromatography-electrospray ionization-low resolution mass spectrometry) analysis revealed a mass consistent with 13^2^-hydroxy-pheophytin a (data not shown), through comparison with spectroscopic data reported on the literature [19,20]. An HR-ESI-MS analysis was performed and showed a monoisotopic *m/z* 887.5697 value for [M + H]^+^ consistent with the molecular formula of C_55_H_74_N_4_O_6_ of 13^2^-hydroxy-pheophytin a (Figure 1). One- and two-dimensional (1D and 2D) NMR experiments and HR-ESI-MS/MS analysis also corroborated this assignment, revealing the typical resonances and correlations of this chlorophyll derivative [19,20].

### 2.2. Isolation and Sctructure Elucidation of Compound **2**

The zebrafish Nile red fat metabolism assay-guided fractionation of the CH_2_Cl_2_/MeOH (dichloromethane/methanol) extract of the cyanobacterium *Nodosilinea* sp. LEGE 06001 yielded compound **2** as a brownish green amorphous solid. The molecular formula was determined as C_50_H_64_N_4_O_6_ on the basis of HR-ESI-MS data, *m/z* 817.4522 [M + H]^+^, which demanded 15 degrees of unsaturation. The ^1^H and ^13^C NMR data for compound **2** showed similarities to what was observed for **1**, indicating that **2** could be a chlorophyll derivative. Comparison of the HR-ESI-MS/MS data between **1** and **2** suggested that compound **2** bears a farnesyl moiety instead of a phytyl group. The difference between the mass of the pseudomolecular ion at *m/z* 817.4522 [M + H]^+^ and the fragment at *m/z* 609.2697 [M + 2H − farnesyl]^+^ (Δ *m/z* 208.1825) is consistent with the presence of a farnesyl substituent as opposed to phytyl (Δ *m/z* 279.36; *m/z* 909.55 [M + Na]^+^ and *m/z* 607.20 [M − phytol]^+^) [18]. Therefore, compound **2** was named 13^2^-hydroxy-pheofarnesin a.

The ^1^H NMR spectrum of **2** (Table 1) showed all the typical resonances of the porphyrin ring: three singlets and an aromatic methyl multiplet (δ_H_ 1.59 m, 3.28 s, 3.44 s, 3.90 s), one methyl triplet (δ_H_ 1.72) and two diastereotopic benzylic proton signals (δ_H_ 3.64, 3.77) corresponding to an aromatic ethyl substituent, three olefinic singlets (δ_H_ 8.71, 9.56, 9.77), a methyl singlet (δ_H_ 3.76) corresponding to a methoxy group, and a vinyl moiety (δ_H_ 6.19, 6.35, 8.03) with the characteristic exomethylene coupling pattern (*J* = 18.7 and *J* = 11.4 Hz). The attached farnesyl moiety was recognized by a large number of overlapping proton signals of aliphatic methylene and methyl functions. Indicative proton resonances were the oxymethylene resonances (δ_H_ 4.44, 4.5 at F1), the vinylic resonances for H-F2 (δ_H_ 5.14), and H_3_-F3^1^ (δ_H_ 1.61), as well as those for the geminal dimethyl at F11^1^ and F12 (δ_H_ 0.85 d).

The ^13^C NMR and 2D NMR data (COSY, HSQC, and HMBC) for **2** allowed the assignment of all carbons on the porphyrin ring and the location of the functional groups, with the exception of the carboxyl group at C-13^1^ (Figure 2 and Appendix A). The connection between the four pyrrole rings was established by the long-range HMBC correlations (Figure 2) of the olefinic protons H-5 to C-4 (ring A), H-10 to C-8 (ring B), C-11, and C-12 (ring C), and H-20 with C-18 (ring D), C-1, and C-2 (ring A). The presence of the methoxycarbonyl group was deduced from the long-range HMBC correlation of the singlet protons of a methyl group resonating at δ_H_ 3.76 H_3_-13^4^ to a carbonyl carbon (δ_C_ 170.8 C-13^3^). This assignment was corroborated by comparison with reported data for similar moieties [19,21,22]. A singlet at δ_H_ 6.11 without an HSQC correlation, but with HMBC correlations to C-13^2^, C-13^3^, and C-15 suggested the presence of a hydroxy group in ring E. Its position at C-13^2^ was deduced by the significant low field shifts of carbons at 13^2^ and 15 (δ_C_ 100.4 and 102, respectively). Moreover, the HMBC data allowed the connection of the farnesyl moiety to the porphyrin system by a long correlation of the methylene group H_2_-F1 (δ_H_ 4.44/4.5) with the carboxyl group at C-17^3^. The assignments from F1 to F12 were confirmed through ^1^H-^1^H TOCSY and ^1^H-^1^H COSY experiments (Figure 2 and Appendix A).

The relative stereochemistry of **2** was confirmed by ROESY experiments and comparison with literature data. Strong ROESY interactions between H-17/H_2_-17^2^ and H_2_-17^2^/H_3_-18^1^ indicated that these protons lay on the same side of the molecule plane (Appendix A). Moreover, the alpha (α) orientation of these protons results from the natural biosynthesis of chlorophylls, in which occurs a stereo-specific reduction of the C-17/C-18 bond of ring D [23]. The stereochemistry of the hydroxy group at C-13^2^ varies naturally to an alpha or beta position (δ_OHβ_ 5.47 to 5.53 in CDCl_3_ [19,24]; δ_OHα_ 5.35 in CDCl_3_ [24]). A correlation of the stereochemistry at C-13^2^ was studied in several pheophytin derivatives comparing the shielding level of the H-17 [25]. It was established that, when the hydroxy group was located on the same side of the molecular plane as H-17 (H-17α and OHα-13^2^), the proton was distinctly deshielded, displaying chemical shifts between δ_H_ 4.54 and 4.69. By contrast, when the hydroxy group had β orientation and the H-17 had α orientation, there was a lack of significant deshielding, and the chemical shift was between δ_H_ 4.13 and 4.29. Thus, despite any strong ROESY signal being observed for this chiral center, the stereochemistry was determined as *S* (OHβ-13^2^) in accordance with these findings. Therefore, the structure of compound **2** was identified as 13^2^(S)-hydroxy-pheofarnesin a.

### 2.3. Lipid-Reducing Activity of **1** and **2**, but not of Chlorophyll a and b

To characterize the lipid-reducing activity of the two isolated chlorophyll derivatives, the zebrafish Nile red fat metabolism assay was used. Exposure of the zebrafish larvae to compounds **1** and **2** resulted in significant neutral lipid-reducing activity in this assay after 48 h (Figure 3A). A significant decrease in Nile red staining was observed for compound **1** at 10, 5, and 2.5 µg/mL, and for compound **2** at 10 and 5 µg/mL (Figure 3B), with half maximal effective concentration (EC_50_) values of 8.9 ± 0.4 µM (7.5 ± 0.3 µg/mL) for **1** and 15.5 ± 1.3 µM (12.7 ± 1.3 µg/mL) for **2**. Toxicity and malformations were evaluated on zebrafish larvae exposed to these two compounds considering general toxicity (death after 24 h or 48 h) and malformations of larval morphological features. No such adverse effects were observed for both compounds. REV (resveratrol) was used as a positive control at a final concentration of 50 µM and significantly reduced the Nile red lipid staining in all bioassays. The solvent control, 0.1% DMSO (dimethyl sulfoxide), did not cause any observable toxicity or malformations toward the zebrafish larvae. Chlorophylls a and b were tested for lipid-reducing activity in this same assay, but neither reduced the fluorescence intensity of Nile red in any of the tested concentrations (156 ng/mL to 10 µg/mL) as shown in Figure 3B. 

### 2.4. Confirmation of Lipid-Reducing Activity in Differentiated 3T3-L1 Spheroids, and Analysis of Lipolysis 

The lipid-reducing activity of **1** and **2** was evaluated in 3T3-L1 spheroids, obtained after a seven-day differentiation period (Appendix A). A significant reduction of lipid accumulation was observed following 48 h of exposure to **1** at 7.5, 15, and 30 µg/mL, and to **2** at 30 µg/mL (Figure 4A). The concentrations were chosen based on the EC_50_ values of compound **1** and **2** in the zebrafish fat metabolism assay. The uptake of this lipids was blocked without affecting cell differentiation, since the impairment of lipid accumulation did not occur in spheroids at the beginning of adipogenesis. No significant reduction of viability of spheroids was observed for exposure to **1** and **2** at 7.5 to 30 µg/mL. Lipases hydrolyze triglycerides into glycerol and free fatty acids making the latter available for cell incorporation [26]. By analysis of the free glycerol content, we observed that compounds **1** and **2** did not induce lipolysis on spheroids of differentiated adipocytes.

### 2.5. qPCR Indicates PPARγ for **1**

In order to study the effects of **1** in lipid metabolism at the transcriptional level, the messenger RNA (mRNA) expression of *fasn*, *mtp*, *pparγ*, and *sirt1* was analyzed in pools of zebrafish larvae, which were exposed to **1** at the EC_50_ concentration for 48h. *Pparγ* mRNA was increased two-fold in response to exposure, while the other genes did not show any significant alteration of their mRNA expression level (Figure 5).

### 2.6. Quantification in Different Source Material of **1**

To understand whether metabolite **1** could be found in biomass suitable for human consumption, the presence of the compound was analyzed by LC-ESI-MS in methanolic extyracts derived from various algae- and plant-based materials. When compared to the strain producing **1**, i.e., *Cyanobium* sp. LEGE 07175 (100%), *Spirulina* sp. biomass contained a slightly higher amount of **1** (120.4%), while lower values were observed for *Chlorella vulgaris* (18.0%), spinach (14.7%), and cabbage (33.0%), as shown in Figure 6. 

## 3. Discussion

Chlorophylls are among the most abundant biological molecules on earth, essential for photosynthesis and ubiquitous in photoautotrophic organisms including cyanobacteria. Chlorophylls are porphyrins, which comprise closed and completely conjugated rings, also referred to as tetrapyrroles. The structure of chlorophyll a, the most widely distributed chlorophyll in nature, features a chelated magnesium atom in the center of tetrapyrrole macrocycle, a characteristic isocyclic fifth ring (ring E) conjoined with ring C, a vinyl group at carbon-3, a ketone at carbon-13^1^, a carbomethoxy group at carbon-13^2^, and a propionic acid moiety at carbon-17 esterified with phytol [27,28,29,30]. Numerous structural alterations occur naturally for chlorophyll a, for example, the formation of magnesium-free pheophorbides and pheophytins, as well as metallo-chlorophyll derivatives. Pheophytin a and pheophorbide a possess some biological properties beneficial for human health including antioxidant properties [31], anti-inflammatory activity [31,32], anti-mutagenic activity [33], cytotoxic effects on cancer cells [34], antiviral activity against hepatitis C virus [35], antimicrobial activity [36], and induction of neuro-differentiation [37]. Regarding metabolic diseases, pheophytin a and pheophorbide a isolated from *Laminaria japonica* inhibited the formation of AGE (advanced glycation end-products) (half maximal inhibitory concentration (IC_50_) 228.71 µM and 49.43 µM, respectively) and of aldose reductase activity (IC_50_ >100 µM and 12.31 µM, respectively) [38]. Reduction of lipid content in differentiated murine 3T3-L1 preadipocytes was shown for a pheophytin (a and b)-rich extract of a plant at 100 µg/mL [39]. 

However, for the herein presented chlorophyll derivative **1**, only a few bioactivities are known. For example, 13^2^(S)-hydroxy-pheophytin a (compound **1**) was shown to have some cytotoxicity (9.9–24.8 µM) on different cancer cells [40]. An anti-proliferative effect on LNCaP cells (prostate cancer) was shown for **1** with an IC_50_ of 20 µM [41]. A methanolic extract from a red alga containing **1** as a minor component amongst seven other compounds (major peaks were unidentified compounds) showed anti-inflammatory activity in vitro at 10 µg/mL [42]. Here, we observed for the first time the lipid-reducing activity of **1** and of the novel compound **2** in the zebrafish Nile red fat metabolism assay. Both effectively reduced the fluorescent staining of neutral lipids with EC_50_ concentrations of 8.9 and 15.5 µM, respectively. Interestingly, neither chlorophyll a nor chlorophyll b showed any activity in that same assay, which highlights the importance of the structural differences found in compounds **1** and **2** for the observed lipid-reducing activity. The zebrafish Nile red fat metabolism assay belongs to the category of small whole animal assays, which have a higher physiological relevance compared to the traditionally used cell assays. Zebrafish larvae were shown to react with known lipid modulator drugs in a similar manner as humans [18]. The zebrafish Nile red fat metabolism assay was already successfully used to identify structurally modified polyphenolic compounds with lipid-reducing activity with EC_50_ values of 0.07–1.67 µM [43], as well as anthraquinone compounds from a marine fungus with EC_50_ values of 0.17–0.95 µM [44]. In comparison to such studies, chlorophyll derivatives **1** and **2** showed lower potency (micromolar range). However, chlorophyll molecules are ubiquitous to photoautotrophic organisms, and may be easy to purify from sources with GRAS status (generally regarded as safe) for human consumption. Here, we quantified the more potent compound **1** by LC–MS/MS in various sources (*Spirulina*, *Chlorella*, cabbage, and spinach) in comparison to our cyanobacterial strain. Compound **1** is produced in a higher quantity in *Spirulina*, but is also present in lower quantity in cabbage, spinach, or *Chlorella*. This result is in line with a study that showed that, amongst chlorophylls, the hydroxy-pheophytins are present in low quantity in different microalgae species [45]. 

The differentiation of murine 3T3-L1 preadipocytes is one of the most used bioassays to identify compounds that interfere with adipogenesis or that act on differentiated adipocytes. In a comparative study utilizing 3T3-L1 adipocytes, zebrafish staining of neutral lipids and the diet-induced obese mice, an effect of chrysophanic acid was detected consistently in all three model systems [46]. Here, we intended to confirm the lipid-reducing activity of **1** and **2** observed for zebrafish larvae in another model system and selected 3T3-L1 preadipocytes. Since 3D cell culture systems present a physiologically relevant alternative to monolayer cell culture, we analyzed the possibility to grow multicellular spheroids (MCS) with 3T3-L1 cells and to induce their differentiation. A similar approach was chosen for immortalized preadipocytes, as well as primary cells from the stromal vascular fractions isolated from adipose tissue from human and mice [47]. Previously, spheroids of 3T3-L1 cells were formed by surface coatings of ELP-PEI (elastin-like polypeptide conjugated to polyethyleneimine), and such spheroids showed higher TG (triglycerides) content and uptake of FA (fatty acids) compared to monolayer cells [48]. The lipid-reducing activity of **1** and **2** was confirmed in MCS from 3T3-L1, as well as the higher potency of **1** that was chosen for the following analyses regarding its molecular mode of action. Toxicity and malformations were evaluated in the same zebrafish whole small animal assay, and, for both **1** and **2**, no general toxicity or malformations were observed. Accordingly, exposure of MCS of differentiated 3T3-L1 did not show any significant viability reduction for both compounds. 

In order to get initial insights into the mechanism of action of **1**, lipolysis was analyzed in the MCS of differentiated 3T3-L1 cells, but such activity was not confirmed. The qPCR analysis of some target genes in zebrafish showed a two-fold induction of *ppar*γ in response to **1**. Peroxisome proliferator activated receptors are involved in the regulation of lipid and carbohydrate metabolism, and PPARγ plays a key role in adipocyte differentiation, lipid storage, and lipogenesis [49]. The yolk cells in zebrafish were shown to process lipids during development, and the exposure of zebrafish with a PPARγ antagonist changed the composition of the lipid profile of 5DPF larvae with increasing/decreasing effects on different lipid species, mainly phospholipids [50]. Studies in mice and clinical trials of PPARγ agonists as thiazolidinediones showed that PPARγ had a role on the distribution of body fat [51]. As a response to TZD (thiazolidinedione) treatment, the metabolically more harmful adipose tissue (the visceral adipose tissue) decreased, while subcutaneous adipose tissue increased, which in part explained the beneficial effects of TZD. Hypothetically, the observed *pparγ* induction in zebrafish in response to **1** may be responsible, to some degree, for the regulation of lipid distribution or alteration of lipid profiles, but our data are not conclusive in this regard. Future studies are needed to identify the mechanism of action of **1**.

Recent studies indicated the existence of absorption and metabolization of oxidized chlorophyll derivatives in both in vitro and in vivo models, establishing a good indication for the future success of compound **1** as a nutraceutical. On the one hand, Chen and Roca demonstrated that oxidized derivatives have a preferential absorption in Caco-2 human intestinal cells (heterogeneous human epithelial colorectal adenocarcinoma cells) over phytylated and other chlorophylls [52]. On the other hand, Vieira et al. highlighted that derivatives with phytyl chains are available for absorption from dietary sources and accumulate to be further metabolized in mice liver [53]. 

## 4. Materials and Methods 

### 4.1. General Experimental Procedures 

One- and two-dimensional (1D and 2D) NMR spectra were acquired with a 400-MHz Bruker Avance III (Bruker, Karlsruhe, Germany) for samples in DMSO-*d*_6_ and with a 600-MHz Bruker Avance III HD frequency for samples in CDCl_3_ and DMSO-*d*_6_ (deuterated dimethyl sulfoxide). Both chemical shifts (^1^H and ^13^C) are expressed in δ (ppm), referenced to the solvent used, and the proton coupling constants *J* are given in hertz (Hz). Spectra were assigned using appropriate COSY, HSQC, HMBC, TOCSY, and ROESY sequences. 

LC-ESI-HRMS analysis was performed on an UltiMate 3000 HPLC (Thermo Fisher Scientific, Waltham, MA, USA) using an ACE Ultracore 2.5 SuperC18 (ACE, United Kingdom), 75 × 2.1 mm inner diameter (id) and the column oven was set to 40 °C. The samples were eluted at 0.35 mL/min over a gradient of 99.5% solvent A (95% H_2_O, 5% MeOH, 0.1% *v*/*v* HCOOH) to reach 10% solvent B (95% isopropanol, 5% MeOH, 0.1% *v*/*v* HCOOH) for 0.5 min, followed by an increase to 60% solvent B for 8 min and by another increase to 90% for 1 min, maintaining those isocratic conditions for over 6 min and returning to the initial conditions for over 1.5 min, before equilibrating for a final 2 min. Analyses were done on a Q Exactive Focus Hybrid Quadrupole Orbitrap Mass Spectrometer (Thermo Fisher Scientific) in the negative and positive ion mode (switching) and controlled by Xcalibur 4.1. The capillary voltage of the heated electrospray ionization (HESI) was set to 3.8 kV. The capillary temperature was 300 °C. The sheath gas and auxiliary gas flow rates were at 35 and 10 (arbitrary units as provided by the software settings).

LC-ESI-HRMS/MS analysis was performed using the same equipment and conditions mentioned previously but with the following gradient: 99.5% solvent A to reach 10% solvent B for 0.5 min, followed by an increase to 60% solvent B for 8 min and by another increase to 90% for 1 min, maintaining those isocratic conditions for over 9 min and returning to the initial conditions for 2 min, before equilibrating for a final 2 min. The UV absorbance of the eluate was monitored at 254 nm, 410 nm, and 665 nm and a full MS scan at the resolution of 70,000 FWHM (full width at half maximum; range of 150–2000 *m*/*z*), and data-dependent MS2 (ddMS2, Discovery mode) at the resolution of 17,500 FWHM (isolation window used was 3.0 amu and normalized collision energy was 35).

Reverse-phase HPLC data were obtained with a Waters Alliance e2695 instrument coupled with a PDA (photodiode array) detector (Mildford, MA, USA) for compound **2** and a Waters 1525 binary pump, coupled to a Waters 2487 detector (monitored wavelengths: 230 and 254 nm) for compound **1**. The software Empower 2 was used for data interpretation. 

Precoated silica gel plates (Merck, KGaA, 60 F-254, 0.5 mm) were used for preparative TLC with visualization under UV light (λ 254 nm).

The phase contrast and red fluorescence images of zebrafish bioassays were obtained with a Leica DM6000B microscope.

### 4.2. Cyanobacterial Growth, Extraction and Fractionation

Initial screening assays led to the selection of two cyanobacterial strains, *Cyanobium* sp. LEGE 07175 and *Nodosilinea* sp. LEGE 06001 isolated from the Portuguese coast [11] and maintained in the LEGEcc in CIIMAR, Matosinhos, Portugal. Strains were cultured in Z8 medium supplemented with marine tropical salt (25 g/L), at 25 °C, with a photoperiod of 14 h/10 h light and dark, respectively, and at a light intensity of 10–30 μmol photons∙m^−2^∙s^−1^. *Cyanobium* sp. LEGE 07175 cultures were grown in 20-L flasks with constant aeration and, at the exponential phase, cells were harvested through centrifugation, before being frozen and freeze-dried. For the *Nodosilinea* sp. LEGE 06001 strain, available freeze-dried biomass was used, which followed the same growth conditions. The biomass of LEGE 07175 (13 g) and LEGE 06001 (56.5 g) was extracted by repeated percolation with a warm mixture of CH_2_Cl_2_/MeOH (2:1, *v*/*v*), yielding a crude extract of 1.9 g and 8.74 g, respectively. Both crude extracts were fractionated by normal-phase (Si gel 60, 0.015–0.040 mm, Merck KGaA, Darmstadt, Germany) VLC (vacuum liquid chromatography) with an increasing polarity grade, from 90% *n*-hex to 100% EtOAc and 100% MeOH [12], giving a total of nine fractions each.

### 4.3. Compound **1** Isolation and Structure Elucidation

VLC fractions E and F (3:2 and 4:1 EtOAc/*n*-Hex, *v*/*v*; ethyl acetate/hexane) showed a lipid-reducing activity with the zebrafish assay; thus, they were pooled and further sub-fractionated by gravity column chromatography using Si gel 60 (0.040–0.063 mm, Merck KGaA, Darmstadt, Germany) as a stationary phase and a gradient of increasing polarity from 1:3 EtOAc/*n*-hex (*v*/*v*) to MeOH producing 12 fractions. The sub-fractions E4 to E6, eluted in 1:3 to 2:3 EtOAc/*n*-hex (*v*/*v*) demonstrated the previous bioactivity, so they were joined. Then, a SPE (solid-phase extraction) separation was performed, using an Si 5g cartridge (Strata SI-1, Phenomenex) and a gradient of 3:2 EtOAc/*n*-hex (*v*/*v*) to MeOH, yielding 14 fractions. Sub-fractions E4E and E4F eluted in 3:2 EtOAc/*n*-hex (*v*/*v*) demonstrated bioactivity and, thus, were submitted to HPLC using a Synergi 4u Fusion (250 × 10 mm 80 Å RP, Phenomenex, Torrance, California, USA) column and using a linear gradient from 7:3 MeCN (aq; acetonitrile) to MeCN for 30 min, before being maintained for 30 min at MeCN (3 mL/min flow). Compound **1** was isolated as the major component at 60 min RT (real-time), yielding 17 mg of pure compound. It was then subjected to LRMS analysis, with a chromatographic column Luna-C18 (250 mm × 4.6 mm, 5 µm, 100 Å, Phenomenex). Samples were eluted at 0.8 mL/min with a gradient starting with 80% H_2_O/MeCN and passing to 100% MeCN in 20 min before maintaining in isocratic conditions for 10 min and returning to the initial conditions for 3 min, followed by 5 min of stabilizing. The monoisotopic mass value 887.33 *m/z* appeared to be the major compound. Through a search of this mass, a hit was found, corresponding to 13^2^-hydroxy-pheophytin a. Then, 1D and 2D NMR analysis, together with HR-ESI-MS and MS/MS analysis, was performed to confirm the hit (Appendix A). 

**13^2^-hydroxy-pheophytin a (1)**: green dark powder, HR-ESI-MS *m/z* 887.5697 [M + H] ^+^, *m/z* 909.5493 [M + Na] ^+^, C_55_H_74_N_4_O_6_. HR-ESI-MS/MS *m/z* 869.5542 [M − OH] ^+^, *m/z* 609.2696 [M − phytol] ^+^, *m/z* 591.2602 [M − OH − phytol]^+^. **^13^C NMR** (101 MHz, DMSO-*d*_6_) δ 192.53 (C-13^1^), 172.88 (C-17^3^), 172.46 (C-13^3^), 168.67 (C-19), 160.59 (C-14), 156.19 (C-16), 153.93 (C-1), 151.07 (C-6), 147.35 (C-3), 147.29 (C-11), 146.71 (C-13), 145.42 (C-9), 143.79 (C-8), 141.79 (C-P3), 138.47 (C-4), 135.18 (C-2), 133.78 (C-7), 130.34 (C-3^1^), 128.08 (C-12), 120.01 (C-3^2^), 118.05 (C-P2), 109.61 (C-15), 106.97 (C-10), 99.65 (C-5), 93.24 (C-20), 89.88 (C-13^2^), 60.40 (C-P1), 52.19 (C-13^4^), 48.81 (C-17), 48.66 (C-18), 36.65-23.65 (C-P4-P14), 29.93 (C-17^1^), 29.90 (C-17^2^), 22.76 (C-18^1^), 22.54 (C-P15^1^), 22.45 (C-P16), 19.58 (C-P7^1^), 19.50 (C-P11^1^), 18.88 (C-8^1^) 17.72 (C-8^2^), 15.84 (C-P3^1^), 12.53 (C-12^1^), 12.25 (C-2^1^), 10.87 (C-7^1^) ppm. 

**^1^****H NMR** (400 MHz, DMSO-*d*_6_) δ 9.70 (1H, *s*, H-10), 9.35 (1H, *s*, H-5), 8.53 (1H, *s*, H-20), 8.14 (1H, *dd*, *J* = 17.7, 11.5 Hz, H-3^1^), 7.24 (1H, *s*, H-13^2^-OH), 6.23 (1H, *dd*, *J* = 17.8, 1.8 Hz, H-3^2^*E*), 6.02 (1H, *dd*, *J* = 11.6, 1.7 Hz, H-3^2^*Z*), 5.06 (1H, m, H-P2), 4.53 (1H, m, H-17), 4.47 (1H, *m*, H-18), 4.37 (2H, *d*, *J* = 7.1 Hz, H-P1), 3.79 (2H, q, *J* = 7.6 Hz, H-8^1^), 3.61 (3H, *s*, H-12^1^), 3.50 (3H, s, H-13^4^), 3.30 (3H, s, H-2^1^), 3.27 (1H, s, H-7^1^), 2.68–1.75 (H-P4-P14), 2.3–1.8 (H-17^1^-H-17^2^), 1.68 (3H, t, *J* = 7.6 Hz, H-8^2^), 1.55 (3H, d, *J* = 7.2 Hz, H-18^1^), 0.82 (6H, d, *J* = 6.6 Hz, H-P15^1^, H-P16), 0.77 (3H, d, *J* = 6.6 Hz, H-P7^1^), 0.72 (3H, d, *J* = 6.6 Hz, H-P11^1^) ppm. 

### 4.4. Compound **2** Isolation and Structure Elucidation

Fraction E (35.27 mg), resulting from the VLC separation and eluted with *n*-hex/EtOAc (2:3, *v*/*v*), had the most interesting activity in the zebrafish bioassay, and was, thus, further fractionated over prepacked normal-phase SPE SiO2 cartridges (55 μm, 70 Å, 2 g, Phenomenex) using CH_2_Cl_2_/MeOH (from 1:0 to 0:1, *v*/*v*), resulting in seven sub-fractions (E1 to E7). Subsequently, sub-fraction E4 eluted with a mixture of CH_2_Cl_2_/MeOH (1:0 and 99:1, *v*/*v*) had the desirable bioactivity level. This sub-fraction was subjected to another round of SPE chromatography using *n*-hex/CH_2_Cl_2_ (from 7:3 to 0:1, *v*/*v*) and CH_2_Cl_2_/MeOH (from 1:0 to 0:1, *v*/*v*), to provide an additional six sub-fractions (E4A to E4F). The active fraction E4C was submitted to further fractionation through reverse-phase HPLC using a Luna-C18 column (250 mm × 10 mm, 10 µm, 100 Å, Phenomenex, Torrance, CA, USA). The elution was done with a mixture of MeCN/CH_2_Cl_2_ (1:1, *v*/*v*) in a flow of 3 mL/min for 25 min with a continuously polar gradient MeCN/CH_2_Cl_2_ (from 3:2 to 1:1, *v*/*v*). This separation afforded three sub-fractions and, according to the result of zebrafish bioassay, fraction E4C11 was further purified by preparative TLC using precoated silica gel plates with a mixture of MeOH/CH_2_Cl_2_ (0:1, *v*/*v*). The final purification yielded 0.3 mg of compound **2**.

**13^2^-hydroxy-pheofarnesin a (2)**: brownish green amorphous solid, HR-ESI-MS *m/z* 817.4522 [M + H]^+^, calculated for C_50_H_64_N_4_O_6_ (816.4825), HR-ESI-MS/MS *m/z* 840.4339 [M + H + Na]^+^, 609.2697 [M + 2H – farnesyl]^+^, 591.2604 [M − OH − farnesyl]^+^, ^13^C and ^1^H NMR (600 MHz, CDCl_3_) data (see Table 1); for NMR and ESI-MS spectra, see Appendix A.

### 4.5. Determination of 13^2^-hydroxy-pheophytin a by LC–MS

For quantification and confirmation of **1**, HRMS and HRMS/MS data of the extracts were acquired on an Accela HPLC fitted with a Synergi C18 column (4 μm, 80 A, 4.6 mm id × 250 mm, Phenomenex, Torrance, CA, USA), coupled to an Accela PDA detector, Accela autosampler, and Accela 600 pump, and to an LTQ OrbitrapTM XL hybrid spectrometer, controlled by LTQ Tune Plus 2.5.5 and Xcalibur 2.1 (Thermo Scientific). Firstly, 20 µL of each sample was injected and samples were eluted over a gradient of 10% solvent H_2_O (0.1% *v*/*v* HCOOH) to reach 100% solvent MeCN (0.1% HCOOH) for 20 min at a flow rate of 0.8 mL/min. The wavelengths were 235, 285, and 650 nm. The LTQ spectrometer was operated in positive ion mode, and the capillary voltage of the electrospray ionization source (ESI) was set to 3.1 kV. The capillary temperature was 300 °C. The sheath gas and auxiliary gas flow rates were at 40 and 10 (arbitrary unit as provided by the software settings). The capillary voltage was 36 V and the tube lens voltage was 85 V. MS data handling software (Xcalibur 2.1.0 QualBrowser software, Thermo Fischer Scientific) was used.

Five different samples were used to assess the production of **1**: *Cyanobium* sp. LEGE07175 was obtained from the LEGEcc (as previously described); *Spirulina* sp. and *Chlorella Vulgaris* were commercially available; leaves of spinach and cabbage were bought from local supermarkets. 

Approximately 1.0 g of powder from each sample was dissolved in 70 mL of MeOH (100%) by stirring for 2h twice at room temperature [54]. The supernatants were collected and dried by a rotatory evaporator to obtain crude extracts. Each extract was then prepared at a concentration of 1 mg/mL for LC–MS, and 0.2 mg/mL for MS/MS in MeOH (100%). A standard of **1** at 16 µg/mL (MeOH 100%) was also prepared for comparison with the samples prior to publication.

Xcalibur 2.1.0 software was used to search for **1** using a reference value interval of 887.50–888.50 *m*/*z*. The PDA chromatogram was analyzed at a wavelength of 428 nm, as it was the maximum of absorption of **1**. Additionally, this software was used to confirm the presence of **1** in the five different samples through comparison of the mass fragmentation pattern (Appendix A).

### 4.6. Zebrafish Nile Red Fat Metabolism Assay

The lipid-reducing activity of compounds was analyzed with the zebrafish Nile red assay as previously described [43,44]. An approval by an ethics committee was not necessary for the presented work, since chosen procedures are not considered animal experimentation according to the EC Directive 86/609/EEC for animal experiments. In brief, zebrafish embryos were raised from one DPF (days post fertilization) in egg water (60 µg/mL marine sea salt dissolved in distilled H_2_O) with 200 µM PTU (1-phenyl-2-thiourea) to inhibit pigmentation. From three DPF to five DPF, zebrafish larvae were exposed to cyanobacterial fractions at a final concentration of 10 µg/mL with daily renewal of water and fractions in a 48-well plate with a density of 6–8 larvae/well (*n* = 6–8). A solvent control (0.1% DMSO) and positive control (REV, resveratrol, final concentration 50 µM) were included in the assay. Neutral lipids were stained with Nile red overnight at the final concentration of 10 ng/mL. For imaging, the larvae were anesthetized with tricaine (MS-222, 0.03%) for 5 min and fluorescence analyzed with a fluorescence microscope (Olympus, BX-41, Hamburg, Germany). Fluorescence intensity was quantified in individual zebrafish larvae by ImageJ (http://rsb.info.nih.gov/ij/index.html). EC_50_ values of purified compounds and of chlorophyll a (chl a) and b (chl b) analytical standards (Sigma-Aldrich) were determined by dose–response curves in further assays by using a dilution series from 156 ng/mL to 10 µg/mL in seven dilution steps.

### 4.7. Differentiated Murine Preadipocytes Grown as Spheroids

Murine preadipocytes 3T3-L1 cell line (ATCC, USA) at 50,000 cells/mL were seeded in an Ultra-Low attachment 96-well plate in Dulbecco’s modified Eagle medium (DMEM, Invitrogen, Carlsbad, CA) supplemented with 10% fetal bovine serum (FBS), 1% penicillin/streptomycin, and 0.1% amphothericin B, and kept under 37 °C at 5% CO_2_ for five days. Differentiation was then induced on the newly formed spheroids of 3T3-L1 with DMEM supplemented with 10 µg/mL insulin (Sigma-Aldrich, St. Louis, USA), 250 nM dexamethasone (Sigma-Aldrich, St. Louis, MO, USA), and 500 µM isobutylmethylxanthine (Sigma-Aldrich) for three days. The differentiation medium was changed to a maintenance medium containing only insulin (10 µg/mL) on DMEM culture medium, for five days. Spheroids were then exposed to **1** and **2** at three different concentrations, 30 µg/mL, 15 µg/mL, and 7.5 µg/mL, over 48 h in two independent assays with six replicates each (*n* = 6). After exposure, spheroids were stained with Calcein AM (Life Technologies, Carlsbad, CA, USA ) 3 µM and Nile red 4 µM (Sigma-Aldrich, St. Louis, USA) for 1h, and then fluorescence was quantified in an Olympus BX41 coupled with an Olympus DP47 camera, and images were analyzed with Cell Profiller software [55] followed by statistical analysis.

### 4.8. Lipolysis

Lipolysis was evaluated by quantifying the free glycerol content present in 50 µL of the medium where 3T3-L1 organoids were exposed to **1** and **2** over 48 h. Then, 80 µL of free glycerol reagent was added (Sigma-Aldrich, St. Louis, USA), and the mixture was incubated at 37 °C for 5 min. Absorbance was recorded at 540 nm. A calibration curve with glycerol standard (0.052 mg/mL) and blank (deionized water) was performed in order to calculate glycerol content (mg/mL) according to the following equation:(A sample − A blank)/(A standard − A blank) × standard concentration.(1)

### 4.9. Real-Time PCR

Zebrafish larvae were exposed to **1** and **2** at EC_50_ concentration and to a solvent control group (DMSO 0.1%) for 48 h between 3DPF and 5DPF as described above. Eight biological replicates consisting of a pool of 10 zebrafish larvae each were sampled per group. The protocol for RNA extraction, quantification, and further processing for real-time PCR analysis followed the one described in Reference [43]. Target gene expression (fatty-acid synthase (FASN), sirtuin 1 (SIRT1), peroxisome proliferator activated receptor gamma (PPARγ), microsomal triglyceride transfer protein (MTP) was normalized to the combination of reference genes (beta-2 microglobulin, B2M/ elongation factor 1-alpha, EF1A). A multiple reference gene approach was chosen for normalization of mRNA expression in order to avoid quantification bias [56]. Real-time PCR was performed using the iQ5 real-time PCR machine (Bio-Rad) and samples were run as described in Reference [57].

### 4.10. Statistics

The data from bioactivity quantifications and mRNA expression are represented as box-whisker plots with values in 5–95 percentiles. The Gaussian distribution was tested by a Kolmogorov–Smirnov normality test (*p* value < 0.05), and homogeneity of variance was determined by Bartlett’s test. One-way ANOVA followed by Dunnett post hoc test (parametric distribution) and Kruskal–Wallis followed by Dunn’s post hoc test (non-parametric distribution) were used to compare the solvent control group (DMSO) and the fractions. Statistically significant differences were considered with *p*-values <0.05.

The data from dose–response curves were used to determine EC_50_ values for bioactivity level. Mean intensity fluorescence data were normalized to the mean values of the solvent control (100%) and to mean values of the 50 μM resveratrol positive control (0%), and concentrations of the compound were log-transformed. A non-linear regression was applied with a variable slope and least square fitting to obtain the dose–response curves. 

## 5. Conclusions

A known and a novel chlorophyll derivative were discovered from marine cyanobacteria with significant lipid-reducing activities in the zebrafish Nile red fat metabolism assay and in MCS of differentiated 3T3-L1 adipocytes. The more potent compound **1** is produced in large quantities in *Spirulina*, which has the GRAS status for human consumption. Lipolysis is not involved in the lipid-reducing activity, but *pparγ* mRNA expression was altered. Regarding its biological properties and sources for production, **1** could be developed as a nutraceutical with lipid reduction activity in the future.

## Figures and Tables

**Figure 1 marinedrugs-17-00229-f001:**
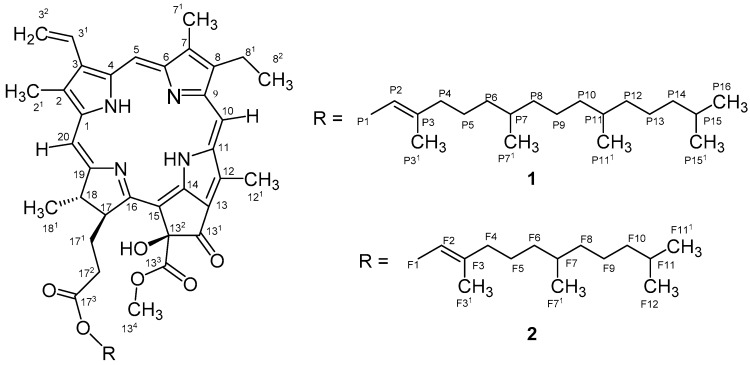
Planar structure of compounds **1** and **2**.

**Figure 2 marinedrugs-17-00229-f002:**
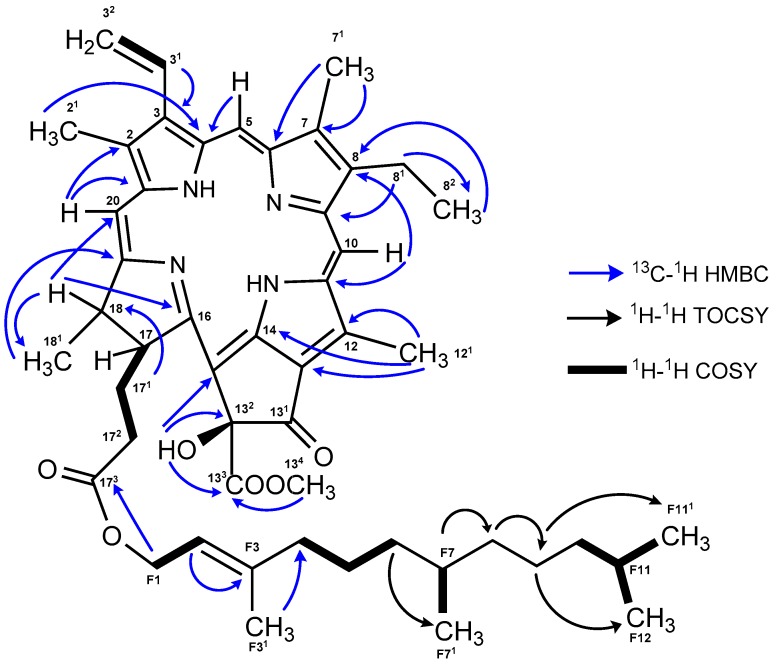
Key ^13^C-^1^H HMBC, ^1^H-^1^H COSY, and ^1^H-^1^H TOCSY correlations for **2**.

**Figure 3 marinedrugs-17-00229-f003:**
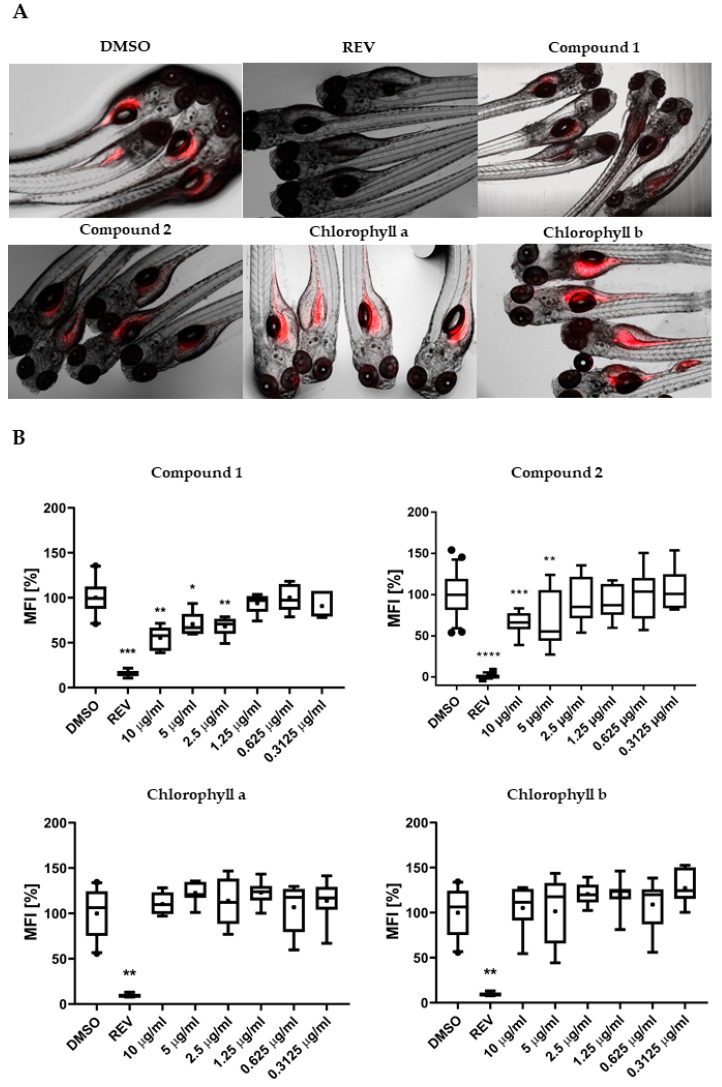
(**A**) Representation of the zebrafish Nile red fat metabolism assay. Strong fluorescence signal is present in zebrafish larvae from the solvent control around the yolk sac and stomach/intestine. Compounds **1** and **2** decreased the Nile red staining, in contrast to chlorophyll a and b. (**B**) Quantification of lipid-reducing activity in the zebrafish Nile red fat metabolism assay after exposure over 48 h. Solvent control was 0.1% dimethyl sulfoxide (DMSO) and positive control was 50 µM resveratrol (REV). Values are expressed as mean fluorescence intensity (MFI) relative to the DMSO group and are derived from six to eight individual larvae per treatment group. The data are represented as box-whisker plots from the fifth to 95th percentiles. Asterisks highlight significant altered fluorescence intensities that indicate changes of neutral lipid level (**** *p* < 0.0001; *** *p* < 0.001; ** *p* < 0.01; * *p* < 0.05).

**Figure 4 marinedrugs-17-00229-f004:**
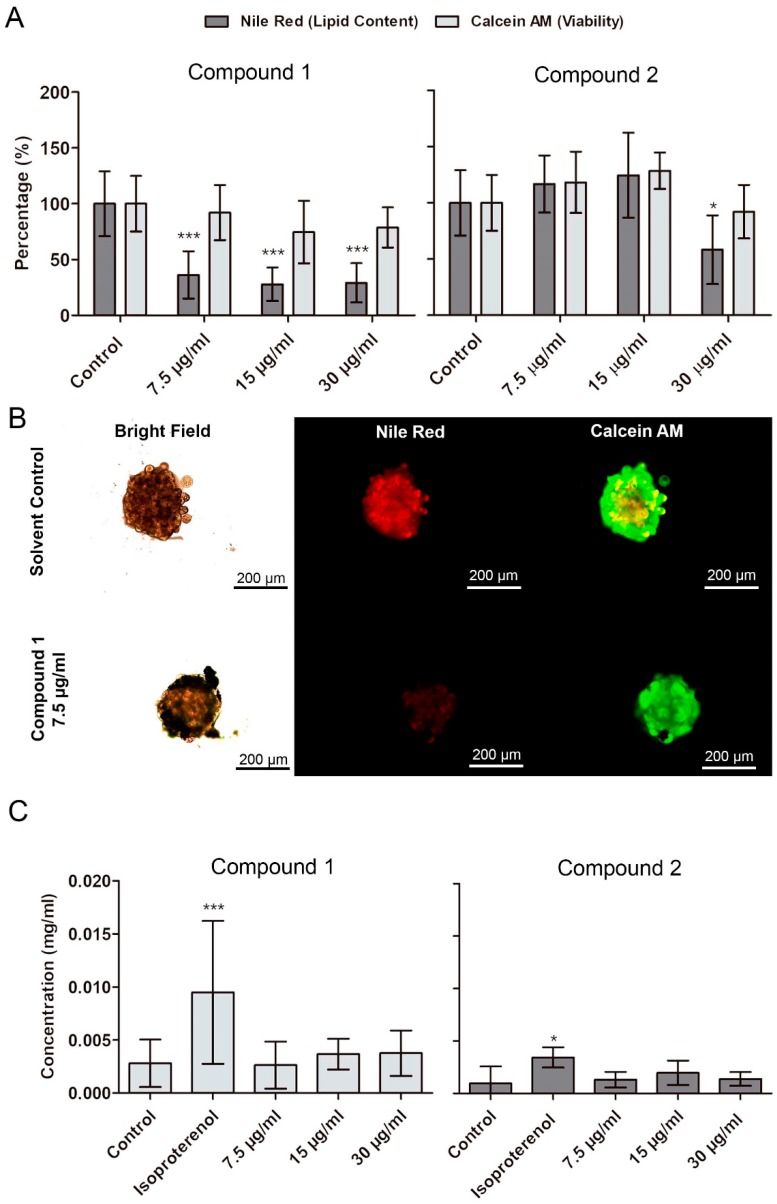
Quantification of lipid content (Nile red) and viability (calcein AM) in differentiated 3T3-L1 spheroids after exposure to **1** and **2** over 48 h. (**A**) Results of quantification of fluorescence by CellProfiler software (mean ± SD). (**B**) Representative images from fluorescence microscopy. Statistical differences to the solvent control were analyzed by one-way ANOVA, followed by a Dunnett’s multiple comparison post-test (*** *p* <0.001, ** *p* < 0.01, * *p* < 0.05). (**C**) Quantification of free glycerol on the medium where 3T3-L1 organoids were exposed to **1** and **2** over 48 h. Data represent means ± SD. No significant alterations on free glycerol content in the medium were observed. Kolmogorov–Smirnov test was used to test normality of the data, followed by a Dunnett’s multiple comparison post-test (*** *p* <0.001, ** *p* < 0.01, * *p* < 0.05).

**Figure 5 marinedrugs-17-00229-f005:**
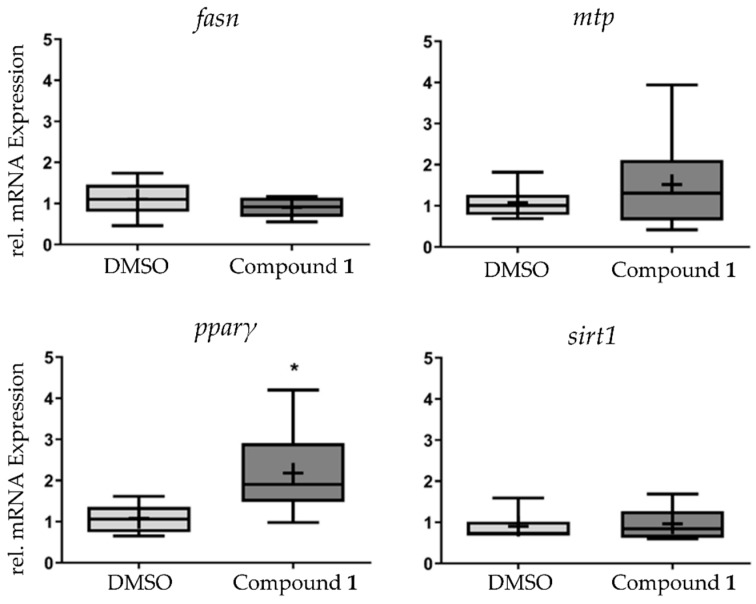
Analysis of messenger RNA (mRNA) expression of *fasn*, *mtp*, *ppar*γ, and *sirt*1 after exposure to **1** at the half maximal effective concentration (EC_50_) for 48 h. Data are presented as box-whisker plots (5–95%) from *n* = 8 replicates, consisting of each replicate from a pool of 10 zebrafish larvae. Significant differences are presented as asterisks, * *p* < 0.05.

**Figure 6 marinedrugs-17-00229-f006:**
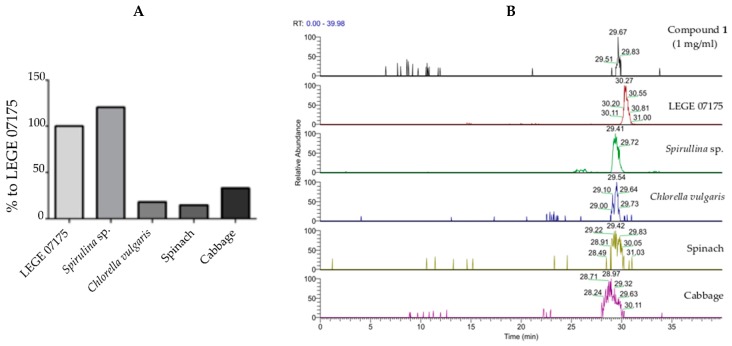
Comparative quantification of **1** in different materials. (**A**) Quantification data are shown as percentage relative to the producing strain of **1**. Samples were prepared at 1 mg/mL in MeOH (100%). (**B**) LC-ESI-MS analysis of the selected samples showing the presence of **1** in all selected materials, compared to the standard (compound **1**) retention time. The peaks selected for analysis had a mass range of *m/z* 887.5593–887.5793 and the following retention times: Compound **1**, 29.67 min; LEGE 07175, 30.27 min; *Spirulina* sp., 29.41 min; *Chlorella vulgaris*, 29.54 min; spinach, 29.42 min; and cabbage, 28.97 min.

**Table 1 marinedrugs-17-00229-t001:** ^1^H and ^13^C (600 MHz) data (δ, ppm) of compound **2** at 600 MHz in CDCl_3_ (deuterated chloroform).

Position	δ_H_ (*J* in Hz)	δ_C_	Position	δ_H_ (*J* in Hz)	δ_C_
1		141.3	15		102.0/100.4 ^2^
2		131.5	16		166.3
2 ^1^	3.44 s	11.5	17	4.07 dd (7.4)	53.1
3		133.4	17 ^1^	2.56 dd (7.7) 1.83 dd (9.1)	30.7
3 ^1^	8.03 dd (11.5, 17.8)	128.3	17 ^2^	2.46 m 2.17 m	31.7
3 ^2^	6.35 d (18.7) 6.19 d (11.4)	122.1	17^3^		173.4
4		135.9	18	4.46 m (7.8)	49.7
5	9.56 s	99.1	18 ^1^	1.59 m	21.7
6		155.7	19		171
7		136.6	20	8.71 s	93.3
7 ^1^	3.28 s	10.8	F1	4.5 m 4.44 m (6.8)	60.8
8		145.6	F2	5.14 t (6.5)	117.1
8 ^1^	3.77 m 3.64 m	19.0	F3		
8 ^2^	1.72 t (7.7)	17.0	F3 ^1^	1.61 s	15.8
9		149.9	F4	1.89 m (7.7)	39.9
10	9.77 s	103.4	F5	1.29 m (7.4)	22-24 ^1^
11		128.8	F6	1.01 m	36.6
12		131.5	F7	1.33 m	32.1
12 ^1^	3.90 s	11.9	F7 ^1^	0.8 m	19.2
13		133.2	F8	2.35 t (7.5)	32.6
13 ^1^		^1^	F9	1.65 m (7.3)	24.3
13 ^2^-OH	6.11 s	110.4/102.0 ^2^	F10	1.11 m	38.9
13^3^		170.8	F11	1.51 m	27.4
13^4^-OCH_3_	3.76 s	53.4	F11 ^1^	0.85 m (6.6)	22.1
14		140.2	F12	0.85 m (6.6)	22.1

^1 13^C spectra signal not detected or very-low-intensity signal. ^2^ Assignment undetermined due lack of specific correlations.

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
