# Peer review of "Chlorophyll Derivatives from Marine Cyanobacteria with Lipid-Reducing Activities"

_marinedrugs, 2019, doi:10.3390/md17040229_

Round 1
Reviewer 1 Report
The work entitled “Chlorophyll derivatives from marine cyanobacteria with lipid reducing activities”, by Freitas and co-workers, describes the isolation of two chlorophyll derivatives with lipid-reducing activity, including a new compound, from marine cyanobacteria, through a bioassay-guided approach.
The experimental work is well organized and the data well-presented. Nevertheless, some issues can be addressed:
General comments:
1. The authors dedicate their survey to the isolation of compounds from natural origin, and with lipid-reducing potential. However, few lines are dedicated to obesity in the introductory part of the manuscript. At least, two or three sentences describing the disorder and the need for new anti-obesity compounds should be added, in order to best fit the objective of the work.
2. Why did the authors select the strains Cyanobium sp. and Nodosilinea sp. for the search for new compounds with lipid-reducing activity?
3. The evaluation of the lipid-reducing capacity of compounds 1 and 2 has been performed through the Zebrafish Nile red assay and in differentiated 3T3-L1 spheroids. In the first model, the authors selected a concentration range of 0.31-10µg/mL, while in the second, they tested concentrations from 7.5-30µg/mL. Can the authors comment these differences?
4. The authors conclude that compound 1, the most potent, could be developed as a nutraceutical with lipid reducing activity. In this regard, can the authors make any comment about the absorption of this compound?
5. Regarding the use of cyanobacteria as source of compound 1, the most potent, do the authors point any advantage when comparing to other terrestrial sources, where it is also present, such as cabbage and spinach?
Minor issues:
Please make a revision of the whole manuscript. There is a lack of punctuation, with emphasis on commas, that should be corrected.
Provide the name of the solvent abbreviations, at their first occurrence in the text, when they are not referred by the molecular formula, e.g., EtOAc, MeCN.
The EC50 values of compounds 1 and 2 are presented in µM, while the graphic representation is in µg/mL. I would suggest the authors to add the conversion to µg to the EC values, e.g. 8.9 ± 0.4 μM (x ± y μg).
Line 27: Please add the meaning of GRAS (generally regarded as safe) in the Abstract section.
Line 36: “Obesity is an increasing epidemic, since a higher percentage of the world’s population is overweight..” Higher than? Do the authors want to refer to the last few years, the last decades? Please rewrite the sentence.
Line 38: “Some natural products are being used in the clinic to treat obesity, as for example orlistat,…” Please rephrase; orlistat is derived from a natural compound, but it cannot be said that it is a natural product used in the clinic.
Line 51: “Some natural products were already successfully isolated…” Do the authors mean “Some natural compounds”?
Lines 60-62: The work by Jones et al. should be better framed in the context, once this way, the sentence seems a bit lost in the context.
Line 69: “Photoautotrophic organisms”, instead of “photoautotrophic organism material…”
Line 86: provide the name of the genus for strain LEGE 06001
Line 427: Please provide the number of independent assays performed for the evaluation of compounds 1 and 2 in the differentiated 3T3-L1 spheroids.
Line 435: DMEM culture medium, instead of “DMEM cultured medium”.
Lines 438-439: …and then fluorescence was quantified in an Olympus BX41, instead of “and then fluorescence was quantified on fluorescent microscopy under an Olympus BX41…”
Line 442: Please rewrite.
Line 443: was collected
Line 449: exposed to
Line 451: …consisting of a pool of 10 zebrafish larvae each, were sampled per group.
Figure 4: Please provide the meaning of the error bars (SD, SEM).
Author Response
The work entitled “Chlorophyll derivatives from marine cyanobacteria with lipid reducing activities”, by Freitas and co-workers, describes the isolation of two chlorophyll derivatives with lipid-reducing activity, including a new compound, from marine cyanobacteria, through a bioassay-guided approach.
The experimental work is well organized and the data well-presented. Nevertheless, some issues can be addressed:
General comments:
1. The authors dedicate their survey to the isolation of compounds from natural origin, and with lipid-reducing potential. However, few lines are dedicated to obesity in the introductory part of the manuscript. At least, two or three sentences describing the disorder and the need for new anti-obesity compounds should be added, in order to best fit the objective of the work.
CIIMAR: We added some sentences to the introduction as requested.
2. Why did the authors select the strains Cyanobium sp. and Nodosilinea sp. for the search for new compounds with lipid-reducing activity?
CIIMAR: We performed initial screenings assays with several strains for bioactivity towards reducing the neutral lipids in zebrafish embryos. These two belonged to the most promising and were selected for the isolation of responsible compounds . We have added a clarification at the M&M section.
3. The evaluation of the lipid-reducing capacity of compounds 1 and 2 has been performed through the Zebrafish Nile red assay and in differentiated 3T3-L1 spheroids. In the first model, the authors selected a concentration range of 0.31-10µg/mL, while in the second, they tested concentrations from 7.5-30µg/mL. Can the authors comment these differences?
CIIMAR: The screening of bioactivity and isolation of responsible compounds was based on the zebrafish assay at concentration of 10 µg/ml. For EC50 values, the concentration ranged from 0.31 to 10 µg/ml. to the second assay (3T3L1 spheroids) were used to confirm this bioactivity in another model. Here, we selected the EC50 values obtained with the zebrafish assay (7.5 µg/ml for 1 and 12.7 µg/ml for 2) and added two higher concentrations. This information was added to the results section.
4. The authors conclude that compound 1, the most potent, could be developed as a nutraceutical with lipid reducing activity. In this regard, can the authors make any comment about the absorption of this compound?
CIIMAR: As requested, we added some studies and findings in the discussion: Line 289-296.
5. Regarding the use of cyanobacteria as source of compound 1, the most potent, do the authors point any advantage when comparing to other terrestrial sources, where it is also present, such as cabbage and spinach?
CIIMAR: The purified hydroxy-pheophytin molecule is the same – independent of its source. However, our quantification in different sources revealed a higher abundance of compound 1 in cyanobacterial sources compared to other photoautotrophic organisms (including terrestrial). Additionally, sustainable large-scale cultures of microalgae are possible.
Minor issues:
Please make a revision of the whole manuscript. There is a lack of punctuation, with emphasis on commas, that should be corrected.
Provide the name of the solvent abbreviations, at their first occurrence in the text, when they are not referred by the molecular formula, e.g., EtOAc, MeCN.
CIIMAR: We have reviewed the manuscript and corrected punctuation and abbreviations as requested.
The EC50 values of compounds 1 and 2 are presented in µM, while the graphic representation is in µg/mL. I would suggest the authors to add the conversion to µg to the EC values, e.g. 8.9 ± 0.4 μM (7.5 ± 0.3 μg/ml).
CIIMAR: Changed as requested in results section.
Line 27: Please add the meaning of GRAS (generally regarded as safe) in the Abstract section.
CIIMAR: Changed as requested.
Line 36: “Obesity is an increasing epidemic, since a higher percentage of the world’s population is overweight..” Higher than? Do the authors want to refer to the last few years, the last decades? Please rewrite the sentence.
CIIMAR: We changed the word into “considerable percentage”.
Line 38: “Some natural products are being used in the clinic to treat obesity, as for example orlistat,…” Please rephrase; orlistat is derived from a natural compound, but it cannot be said that it is a natural product used in the clinic.
CIIMAR: Changed as requested.
Line 51: “Some natural products were already successfully isolated…” Do the authors mean “Some natural compounds”?
CIIMAR: We corrected “natural products” into “natural compounds”.
Lines 60-62: The work by Jones et al. should be better framed in the context, once this way, the sentence seems a bit lost in the context.
CIIMAR: Corrected as requested.
Line 69: “Photoautotrophic organisms”, instead of “photoautotrophic organism material…”
CIIMAR: Changed as requested.
Line 86: provide the name of the genus for strain LEGE 06001
CIIMAR: Changed as requested.
Line 427: Please provide the number of independent assays performed for the evaluation of compounds 1 and 2 in the differentiated 3T3-L1 spheroids.
CIIMAR: Information was added as requested.
Line 435: DMEM culture medium, instead of “DMEM cultured medium”.
CIIMAR: Changed as requested.
Lines 438-439: …and then fluorescence was quantified in an Olympus BX41, instead of “and then fluorescence was quantified on fluorescent microscopy under an Olympus BX41…”
CIIMAR: Changed as requested
Line 442: Please rewrite.
CIIMAR: We have rewritten the sentence for clarity.
Line 443: was collected
CIIMAR: Changed as requested.
Line 449: exposed to
CIIMAR: Changed as requested.
Line 451: …consisting of a pool of 10 zebrafish larvae each, were sampled per group.
CIIMAR: Changed as requested.
Figure 4: Please provide the meaning of the error bars (SD, SEM).
CIIMAR: Information added as requested.
Reviewer 2 Report
In this study, the authors investigated the new cyanobacterial compounds with lipid reducing activity using the zebrafish Nile red fat metabolism assay. The authors concluded that known and novel chlorophyll derivatives were discovered from marine cyanobacteria with relevant lipid reducing activities, which in the future may be developed into nutraceuticals.
Comments
This is an interesting study. The manuscript is well-written. The reviewer has some minor concerns as follows:
1. In discussion section, the authors mentioned that “Toxicity and malformations were evaluated in the same zebrafish whole small animal assay, and for both 1 and 2, no general toxicity”. This should be described in detail. The authors may provide these toxicity data in the results section.
2. In Figure 4B, what is the HPA 7.5 μg/ml?
3. page 10, line 271, “ppary” changes to “pparγ” (ppar-gamma).
Author Response
In this study, the authors investigated the new cyanobacterial compounds with lipid reducing activity using the zebrafish Nile red fat metabolism assay. The authors concluded that known and novel chlorophyll derivatives were discovered from marine cyanobacteria with relevant lipid reducing activities, which in the future may be developed into nutraceuticals.
Comments
This is an interesting study. The manuscript is well-written. The reviewer has some minor concerns as follows:
1. In discussion section, the authors mentioned that “Toxicity and malformations were evaluated in the same zebrafish whole small animal assay, and for both 1 and 2, no general toxicity”. This should be described in detail. The authors may provide these toxicity data in the results section.
CIIMAR: Information was added as requested.
2. In Figure 4B, what is the HPA 7.5 μg/ml?
CIIMAR: In Figure 4B it should say compound 1 7.5 ug/ml. We changed this information correspondingly.
3. page 10, line 271, “ppary” changes to “pparγ” (ppar-gamma).
CIIMAR: Changed as requested.